# Hematology, Ultrastructure and Morphology of Blood Cells in Rufous-Winged Buzzards (*Butastur liventer*) from Thailand

**DOI:** 10.3390/ani12151988

**Published:** 2022-08-05

**Authors:** Pornchai Pornpanom, Chaiyan Kasorndorkbua, Preeda Lertwatcharasalakul, Chaleow Salakij

**Affiliations:** 1Akkhararatchakumari Veterinary College, Walailak University, Nakhon Si Thammarat 80160, Thailand; 2Department of Pathology, Faculty of Veterinary Medicine, Kasetsart University, Bangkok 10900, Thailand; 3Kasetsart University Raptor Rehabilitation Unit, Faculty of Veterinary Medicine, Kasetsart University, Nakhon Pathom 73140, Thailand

**Keywords:** blood cell, *Butastur liventer*, hematology, KURRU, ultrastructure

## Abstract

**Simple Summary:**

Complete blood counts (CBCs) are a manual tool used for the screening of raptor health. These tests require knowledge of blood cell morphology. This study aims to describe the quantitative and qualitative information of blood cells in rufous-winged buzzards (RWB) admitted into the rehabilitation center. The CBCs of 12 out of 17 RWBs were included for descriptive hematologic values. Heterophils were the most prevalent white blood cells in RWBs. One non-parasitized RWB showed hypochromic erythrocytes with PCV 0.18 L/L, which indicated that anemia in RWBs resulted from non-parasitic causes. The blood cells in RWBs are similar to those in other diurnal raptors, except for the lymphocytes. The electron micrographs confirmed the photomicrograph. This report can be used for the baseline information for rehabilitation of the RWBs in Thailand.

**Abstract:**

In attempt to treat injured raptors and promote conservation awareness, the Kasetsart University Raptor Rehabilitation Unit (KURRU) was established in 2007. The complete blood counts (CBCs) are a manual tool used for the screening of raptor health. These tests require knowledge of blood cell morphology. This study aimed to describe the preliminary information of the hematology, ultrastructure, and morphology of blood cells in rufous-winged buzzards (RWB). There were 17 RWBs admitted into the KURRU. CBCs were manually performed by veterinary technicians. The morphology and morphometry of blood cells were observed from Wright-stained blood smears. Ultrastructure was observed from uranyl acetate and lead citrate-stained sections. The hematologic values were analyzed and described from individual RWBs that were clinically healthy, negative for blood parasites, and had PCV > 0.30 L/L. Consequently, CBCs of 12 out of 17 RWBs were included for descriptive hematologic values. Heterophils were the most prevalent white blood cells in RWBs. Of these 17 RWBs, 1 non-parasitized RWB showed hypochromic erythrocytes with PCV 0.18 L/L, which indicated that anemia in RWBs resulted from non-parasitic causes. The morphology of blood cells in RWBs was similar to those in other diurnal raptors, except that the lymphocytes showed pale or colorless cytoplasm. The electron micrographs highlighted that the basophil contained two types of granules: homogeneous electron-dense granules and reticulated electron-dense granules. The photomicrographs in this report are the scientific reference for identification of blood cells in RWBs. The CBCs from non-parasitized RWBs (clinically healthy) can be used as a cage mate reference in the KURRU. Additionally, we found evidence that evaluations of blood smears together with CBC examination were important in raptors.

## 1. Introduction

Both diurnal and nocturnal raptors (order: Accipitriformes, Falconiformes, and Strigiformes) are apex predators of the food chain, used as a bioindicator of an ecosystem [1,2]. These birds may be a potential animal model for monitoring the environmental contamination of zoonoses or antimicrobial resistance, as those have been reported previously [3,4]. Rufous-winged buzzards (*Butastur liventer*, RWBs) are the resident bird of Thailand and are lawfully protected [5]. It is listed as a least-concern species on the 2022 IUCN red list, and the world population of mature individuals is decreasing [6]. 

In an attempt to contribute public awareness of conservation and rehabilitate the raptors (injured and offspring birds), the Kasetsart University Raptor Rehabilitation Unit (KURRU) was established in 2007 [7]. The rehabilitation procedures include physical and laboratory examination, behavior monitoring, and then release back into suitable habitats [8]. Complete blood counts (CBCs) are routine laboratory tests. Evaluations of CBCs in raptors are performed by using the manual method because using an automated hematologic analyzer is not possible. The reason is that the naked nuclei of lysed red blood cells, thrombocytes, and lymphocytes are all similar in size, which interferes with the use of an impedance-based automated analyzer [9,10]. 

The manual evaluation of CBCs in raptors requires knowledge of blood cell morphology. This is for the identification of blood cells during the step of the differential count of white blood cells. Although there are reports of blood cell morphology in some raptor species having no significant differences [11,12], there is a report of different characteristics for eosinophils’ granules among nocturnal raptors [1]. Therefore, reports of blood cell characteristics in each avian species are important, and this will provide scientific evidence for laboratory work and further research. To the best of our knowledge, there are some descriptions of blood cell characteristics in raptorial birds [1,11,12,13,14,15,16] but there are no reports of the quantitative and qualitative characteristics of blood cells in RWBs.

Thus, the authors aim to describe the baseline information of blood cell characteristics in RWBs from Thailand, with an emphasis on morphology, morphometry, and ultrastructure. This article also reports the mean and median hematologic values of clinically healthy raptors. These are the valuable information supporting the rehabilitation of RWBs, which are decreasing in number. Additionally, this report can be the guideline for identification of blood cells in RWBs. Although during the long 7-year period of sample collection, there have not been many RWBs admitted to the KURRU, the small sample size, which is albeit not sufficient for the calculation of reference intervals, can be used to report hematologic values [17,18], especially in wildlife animals for which sample collection is likely difficult and whose population is also small.

## 2. Materials and Methods

### 2.1. Sample Collection and Managements

During September 2012 and September 2019, 17 rufous-winged buzzards (RWBs) were admitted to the Kasetsart University Raptor Rehabilitation Unit, Faculty of Veterinary Medicine, Kasetsart University, Kamphaeng Saen, Nakhon Pathom, Thailand (14°1′N, 99°58′E). These buzzards had 1 mL of blood opportunistically collected from their right jugular vein. Then, the blood was transferred into an EDTA-containing tube, kept in an ice-box, and submitted to the laboratory for further analysis. The distance from the KURRU was 200 m. Blood samples were then observed for blood parasites using molecular techniques or combined molecular and microscopic techniques, as described in the previous articles [7,19,20]. There were two species of blood parasites found in RWBs, including *Haemoproteus* sp. and *Plasmodium* sp. [7], and the data were shown (Table 1). The genus of blood parasites was identified based on microscopic and molecular characteristics.

### 2.2. Hematologic Values

Hematologic values were manually evaluated by veterinary technicians following the standard methods from the previous reports [11,12], during a period of other projects [7,19,20]. Briefly, the packed cell volume (PCV) was determined by the hematocrit centrifuge method (5 min at 10,000 rpm). The hemoglobin (Hb) concentration was measured by the cyanmethemoglobin method after the removal of naked nuclei by centrifugation. The total red blood cell (RBC) and white blood cell (WBC) counts were manually determined using a hemocytometer after dilution with Natt and Herrick’s solution (1:200). The RBC indices, mean cell volume (MCV), mean corpuscular hemoglobin concentration (MCHC), and mean corpuscular hemoglobin (MCH) were calculated by using the following formula: MCV (fL) = PCV (%) × 10 / RBCs (10^12^ cells/L); MCHC (g/dL) = Hb (g/L) × 100 / PCV (%); and MCH (pg) = Hb (g/L) × 10 / RBCs (10^12^ cells/L). WBC differential count was performed on the Wright-stained blood smears. The number of thrombocytes was simultaneously counted with WBC differential count and reported as the number per 100 WBCs. The percentages of reticulocytes were observed from new methylene-blue-stained blood smears. 

Total solid was measured using a refractometer after the plasma was separated from other blood components by capillary centrifugation (5 min at 10,000 rpm). The level of fibrinogen was measured using a precipitation method. According to the American Society for Veterinary Clinical Pathology (ASVCP) guidelines for determination of *de novo* reference intervals (RIs) [18], reference samples lower than 20 cannot be reported as RIs. Nevertheless, the mean, median, minimum, and maximum hematologic values can be described as in a previous report [17]. 

### 2.3. Blood Cell Morphology and Morphometry

Blood cell morphology was observed from Wright-stained blood smears (in-house preparation using Wright’s eosin methylene blue, Merck KgaA, Darmstadt, Germany). Measurements of blood cell morphometry were performed in 5 non-parasitized RWBs. The number of measured cells was 50 cells of RBCs (length, width, and area) and 30 cells of each WBC type (diameter). The Nikon ECLIPSE C*i*-L (Nikon, Tokyo, Japan), equipped with a Nikon DS-F*i*3 digital camera, and the NIS Elements D imaging software (Nikon, Tokyo, Japan) were used for the analysis of blood cells.

### 2.4. Transmission Electron Microscopy

The EDTA blood was processed for transmission electron microscopy (TEM) as per previous reports [11,12]. In brief, buffy coats from centrifuged capillary tubes (5 capillary tubes) were fixed in 2.5% glutaraldehyde in 0.1 M phosphate buffer at 4 °C for 24 h. Then, the fixed buffy coats were processed post-fixed with 1% osmium tetroxide for two hours and embedded in Spurr’s epoxy resin. Ultrathin sectioning was performed, stained with uranyl acetate and lead citrate. The stained sections were used for observation of the ultrastructure using the JEM1230 transmission electron microscope (JEOL, Tokyo, Japan) or the HT770 transmission electron microscope (Hitachi, Tokyo, Japan). This depended on the availability of the transmission electron microscope. Identification of blood cells was based on the number, shape, distribution of granules, and the nuclear appearance.

### 2.5. Statistical Analysis

Descriptive statistics (mean, median, min, and max) were used to describe the hematologic values and morphometry of blood cells of non-parasitized rufous-winged buzzards. The hematologic values of non-parasitized raptors that had PCVs higher than 0.30 L/L were used for descriptive hematologic values.

## 3. Results

Seventeen admitted rufous-winged buzzards (RWBs) were admitted to the Kasetsart University Raptor Rehabilitation Unit. Blood parasites were found in 3 out of the 17 RWBs, including *Haemoproteus* sp. and *Plasmodium* sp. (Table 1). Furthermore, two non-parasitized samples showed low PCV values (< 30 L/L), which were 0.18 and 0.26 L/L. Additionally, the non-parasitized RWB with a PCV of 0.18 had hypochromic erythrocytes (Figure 1). Three parasitized and two non-parasitized samples with low PCV were excluded from the report of descriptive hematologic values.

### 3.1. Hematologic Values

The total 12 non-parasitized samples that had PCV values higher than 0.30 L/L (clinically healthy) were reported for hematologic values (Table 2). The PCV of clinically healthy RWBs was 0.32–0.46 L/L. Hemoglobin concentration was 92.0–153.0 g/L. Total red blood cell count was 1.31–3.08 (×10^12^/L). Total white blood cell count was 8.23–20.37 (×10^9^/L). Heterophils were the most prevalent white blood cells (WBCs) and ranged from 2.65 to 11.61 cells × 10^9^/L. Lymphocytes were the second most prevalent WBCs and ranged from 1.39 to 5.74 cells × 10^9^/L. Thrombocytes in clinically healthy RWBs were 165–347 cells per 100 WBCs. The punctate and aggregate reticulocytes in peripheral blood circulation were 9.0–69.0 % and 3.6–33.0%, respectively. The total solids ranged from 40.0 to 56.0 g/L. The fibrinogen ranged from 1.0 to 4.0 g/L.

### 3.2. Blood Cell Characteristics

The red blood cells or erythrocytes of RWBs were oval in shape with an oval nucleus that was condensed chromatin. Their cytoplasm contained homogeneously dark-red staining. The average length, width, and area were 12.21 ± 0.6 µm, 6.95 ± 0.4 µm, and 71.11 ± 5.5 µm^2^, respectively (Table 3). Heterophils (diameter: 11.17 ± 0.6 µm) were round in shape containing a segmented nucleus and dark eosinophilic spindle-shaped granules (Figure 2A). Eosinophils (diameter: 11.15 ± 0.6 µm) were round in shape, containing bright eosinophilic round-shaped granules (Figure 2B). Basophils (diameter: 9.06 ± 1.0 µm) were round in shape with a round nucleus that was obscured by small round basophilic granules (Figure 2C). 

The lymphocytes (diameter: 8.00 ± 0.9 µm) in RWBs were round in shape, containing an eccentric nucleus. The cytoplasms of lymphocytes were very pale blue, and sometimes, the cytoplasm was bright blue. Lymphocytes with azurophilic granules can be found in clinically healthy RWBs (Figure 2D). Monocytes (diameter: 12.66 ± 0.8 µm) in these raptors contained round- or irregular-shaped nuclei. The cytoplasm was abundant, containing vacuoles and fine pink materials. (Figure 2E). Thrombocytes were round in shape with a round nucleus. Their cytoplasm was colorless, containing a vacuole, and azurophilic granules can also be found (Figure 2F).

### 3.3. Ultrastructure of Blood Cells

The ultrastructure of the blood cells in RWBs was shown at the end (Figure 3), except for lymphocytes. Electron micrographs revealed the details of granules of heterophils, eosinophils, and basophils (Figure 3A–C). Heterophil granules were spindle-shaped with homogeneous electron density, whereas eosinophils contained round homogeneous electron-dense granules. Basophils contained two types of granules: homogeneous electron dense-granules and reticulated electron dense-granules. Monocytes showed abundant cytoplasm with long cytoplasmic processes, while mitochondria and a rough endoplasmic reticulum were seen (Figure 3D–E). Thrombocytes contained a perinuclear halo that was a major characteristic of these cells (Figure 3F).

## 4. Discussion

The PCVs of 12 clinically healthy rufous-winged buzzards (RWBs) had PCV values ranging from 0.32 to 0.46 L/L. In general, avian species with PCVs lower than 0.35 L/L were considered to be anemic [21], but there was no consensus PCV range in RWBs. Therefore, the PCV range from these clinically healthy non-parasitized RWBs (without hypochromic erythrocytes found in blood smears) can be used as a cage mate reference of hematologic values in Thai RWBs. Nevertheless, a further investigation that gathers more reference individuals might provide a solid reference interval (RI).

Regarding the hemogram change in parasitized RWBs, the PCV in three parasitized RWBs were 0.22, 0.39, and 0.40 L/L (Table 1). Two out of these RWBs showed normal PCV (Table 2), and cage mate referencing indicated that anemia was not always shown in hemosporidian-infected RWBs. This was similar to the description from Salakij et al. (2018) [22] that the *Haemoproteus* did not always result in clinical disease, but sometimes this parasite caused severe anemia (hemolytic anemia) in Harris hawks. However, interpretation of hematologic values from only three infected cases should be performed carefully. In these cases, the level of PCV might be influenced by compensatory mechanism and hydration status.

One anemic non-parasitized RWB (PCV < 0.18 L/L) showed hypochromic erythrocytes (Figure 1). This indicated that anemia in RWBs can develop from non-parasitic causes, such as toxicants, iron deficiency, and inflammation [23]. This might be the evidence that evaluation of blood smear together with CBC examination were important tools for the health screening of RWBs or other raptors. This will reveal the underlying pathological stage or confirm changes in hemogram.

Since the sample size was lower than 20, calculation of RIs is not acceptable [18]. However, a sample size of more than 10 can be used to report mean, median, min, and max values [17]. In this study, sample collection was a drawback. The reason was that the blood sample was opportunistically collected from the admitted RWBs, and the submission of RWBs depends on the authorities and people. Notwithstanding, the RWBs were wild birds and their global number was decreasing [6], together with the long period of sample collection. Therefore, this report is worthwhile. The authors suggested that sample collection from natural habitats with a bigger sample size in future study might be necessary for calculation of hematologic RIs in this raptor or others.

Blood cells of RWBs include erythrocytes (RBCs), leukocytes (WBCs), and thrombocytes. WBCs can be divided into heterophils, eosinophils, basophils, lymphocytes, and monocytes that were similar to other raptors [11,12,14,15,17,18,24,25]. Heterophils were the most prevalent WBCs in RWBs, whereas lymphocytes were the second most prevalent WBCs (Table 2). The appearance of RBCs, granulocytes, and monocytes were quite similar to those in black-shouldered kites (*Elanus caeruleus*), Brahminy kites (*Haliatur indus*), black kites (*Milvus migrans govinda*) [11], Blyth’s hawk eagles (*Nisaetus alboniger*) [24], crested serpent eagles (*Spilornis cheela*), and shikra (*Accipiter badius*) [12], except for lymphocytes. Lymphocytes showed pale or colorless cytoplasm. For thrombocytes, these cells showed variation in shapes and staining patterns. 

Eosinophils of diurnal raptors contained bright eosinophilic round-shaped granules [11,12,24]. This was similar to those in nocturnal raptors [1], except in barn owls and great horned owls. The heterophils of RWBs had no central bodies, which was found in nocturnal raptors [1] and grater adjutants [26]. For other WBCs, basophils contained basophilic granules that usually obscured its nucleus. Lymphocytes contained an eccentric round nucleus and sometimes showed azurophilic granules. Monocytes had abundant cytoplasm that sometimes showed the vacuoles. These were similar to those in other diurnal raptors [11,12,24].

Transmission electron microscopy revealed the fine structure of granules of heterophils, eosinophils, and basophils of RWBs (Figure 3), which was similar to those in other kites [11]. Basophils contained the homogeneous and reticulated electron-dense granules. This electron micrograph was similar to those in other raptors, such as kites [11], crested serpent eagles, shikra [12], and Blyth’s hawk-eagles [24]. The fine structure of thrombocytes showed distinguished peri-nuclear halo and azurophilic granules (Figure 3F) that confirmed the microscopic characteristics (Figure 2).

## 5. Conclusions

This report contained information on hematologic values that can be used for the health screening of rufous-winged buzzards. Additionally, the morphology of blood cells can be used as color guideline pictures for identification of blood cells in rufous-winged buzzards. This report was considered as a preliminary report and the baseline information for further research.

## Figures and Tables

**Figure 1 animals-12-01988-f001:**
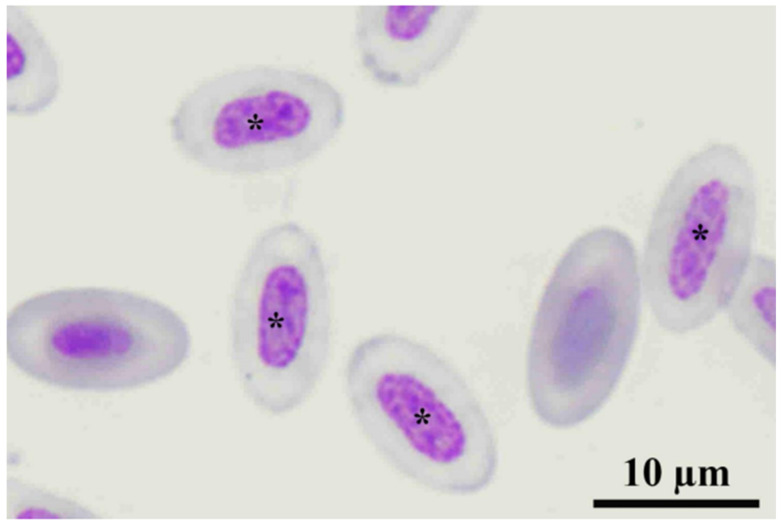
Hypochromic erythrocytes (*) found in rufous-winged buzzard with PCV 0.18 L/L and negative for blood parasites. **Note**, it is not clear if the 2 non-labeled erythrocytes are normochromic or polychromatophilic erythrocytes.

**Figure 2 animals-12-01988-f002:**
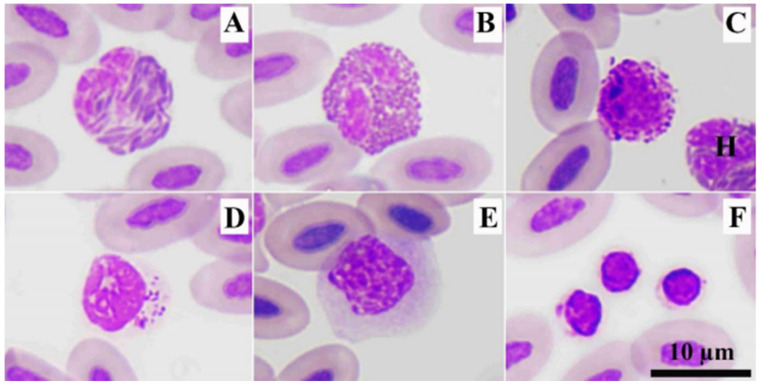
Blood cells of rufous-winged buzzard. Heterophil (**A**), eosinophil (**B**), basophil and heterophil (H) (**C**), lymphocyte with azurophilic granules (**D**), monocyte (**E**), and three thrombocytes (**F**). Wright’s stain.

**Figure 3 animals-12-01988-f003:**
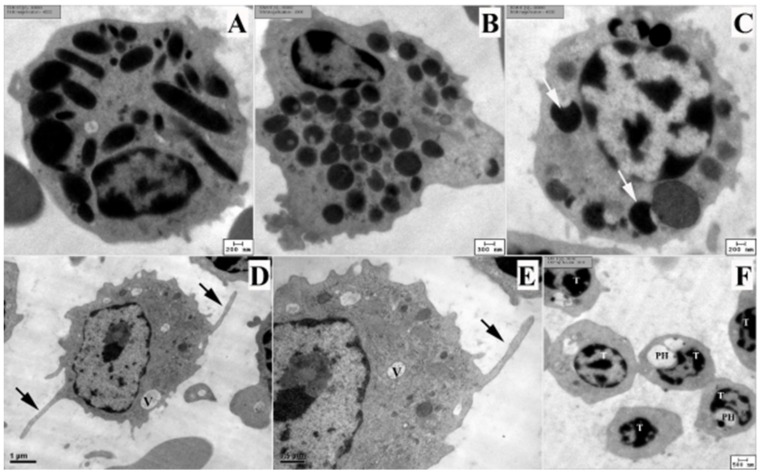
Transmission electron micrographs of blood cells in rufous-winged buzzards. Heterophil (**A**), eosinophil (**B**), basophil (**C**), monocytes (**D**–**E**), and six thrombocytes (**F**). Basophil shows two types of granules: heterogeneous electron dense granules and homogeneous electron dense granules (white arrows). Monocytes show the long cytoplasmic processes (back arrows) and cytoplasmic vacuoles (V). High magnification of mitochondria and rough endoplasmic reticulum (**E**). Some of thrombocytes (T) shows the perinuclear halo (PH). Uranyl acetate and lead citrate stains. The electron micrographs were taken by the HT770 TEM (**A**–**C**,**F**) and JEM1230 TEM (**D–E**).

**Table 1 animals-12-01988-t001:** Summary of blood parasites in rufous-winged buzzards and some hemograms in parasitized birds.

Blood Parasites	KURRU Code ^a^	GenBank No.	PCV (L/L)	Hb (g/L)	RBC (10^12^/L)
*Haemoproteus* sp. ^b^	R14	MZ502239	0.22	7.2	1.54
	KU306	MZ502240	0.39	12.5	2.61
*Plasmodium* sp. ^b^	KU410	MZ502241	0.40	9.4	3.52

^a^ Code is provided by the Kasetsart University Raptor Rehabilitation Unit. ^b^ Reported by Pornpanom et al. (2021).

**Table 2 animals-12-01988-t002:** Descriptive summary of hematologic values of rufous-winged buzzards of both sexes and different ages.

Analytes	Unit	Mean	SD	Median	Min	Max
**Rufous-winged buzzard (*n* = 12)**						
PCV	L/L	0.39	0.0	0.38	0.32	0.46
Hb	g/L	124.6	16.8	125.0	92.0	153.0
RBC	10^12^/L	2.31	0.6	2.34	1.31	3.08
MCV	fL	168	49.7	147	106	275
MCH	pg	58	20.3	50	34	99
MCHC	g/dL	34.4	3.8	33.7	27.9	42.0
WBC	10^9^/L	14.27	3.5	14.06	8.23	20.37
Heterophils	10^9^/L	6.63	3.2	6.31	2.65	11.61
	%	45	14.3	46	18	64
Eosinophils	10^9^/L	1.98	1.2	1.53	0.74	4.86
	%	14	8.8	11	6	33
Basophils	10^9^/L	0.36	0.3	0.34	0.00	1.21
	%	3	2.1	2	0	7
Lymphocytes	10^9^/L	3.55	1.4	3.36	1.39	5.74
	%	26	10.0	25	8	45
Monocytes	10^9^/L	1.76	0.8	1.77	0.38	3.13
	%	13	5.6	12	3	21
Thrombocytes	/100 WBCs	248	63.7	240	165	347
Reticulocytes						
Punctate	%	42.8	26.4	58.0	9.0	69.0
Aggregate	%	19.2	9.3	18.9	3.6	33.0
TS	g/L	49.2	5.3	50.0	40.0	56.0
Fibrinogen	g/L	2.6	1.3	2.0	1.0	4.0

**Table 3 animals-12-01988-t003:** Blood cells’ morphometry in non-parasitized rufous-winged buzzards.

Blood Cells	Mean ± SD	Min–Max
**Red blood cells (250 cells)**		
Length (µm)	12.21 ± 0.6	10.20–14.13
Width (µm)	6.95 ± 0.4	5.96–8.25
Area (µm^2^)	71.11 ± 5.5	57.75–88.23
**White blood cells diameter (150 cells)**		
Heterophils (µm)	11.17 ± 0.6	9.69–12.7
Eosinophils (µm)	11.15 ± 0.6	10.00–12.48
Basophils (µm)	9.06 ± 1.0	6.45–11.23
Lymphocytes (µm)	8.00 ± 0.9	5.59–9.93
Monocytes (µm)	12.66 ± 0.8	10.93–14.68

## Data Availability

Data sharing not applicable.

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
