# Peer review of "Hematology, Ultrastructure and Morphology of Blood Cells in Rufous-Winged Buzzards (*Butastur liventer*) from Thailand"

_animals, 2022, doi:10.3390/ani12151988_

Round 1

Reviewer 1 Report

The manuscript is well written but some further corrections or information is necessary to improve the manuscript.

Abstract:

Line 16: Please add an explanation for the abbreviation RWB in the previous sentence.

Introduction:

Line 34-38: Please check whether the cited literature is adequate for the statements of the first two sentences.

Line 49: New machines are available which evaluate the blood smears, but these also have problems, especially with the differentiation of eosinophils in birds.

Lien 54: Falcons for example have a different morphology of the eosinophils in comparison to other raptor species.

Line 66: Please add: " Which is however not sufficient for the calculation of reference intervals."

Material and methods:

Line 75: After which time were the blood smears prepared and how long was the transport time of the blood to the laboratory and the start of the blood analysis?

Line 84-102: Please add more information about the hemtology methods used or add references.

Line 106: How were the 5 animals selected? Age, sex?

Line 125: How was this cut-off determined?

Results:

Line 128-135: The first paragraph of the results should be moved to the Materials and Methods section, as some of this is also a repetition.

Line 170: Lymphocytes with granulation were found in all healthy RWBs?

Figures:

Is it possible to exchange the figure of the lymphocyte against one without granulation or is the granulation typical in this avian species?

Discussion:

Line 212: Please include more other potential reasons for anemia.

Line 242-250: The last paragraph of the discussion could be shorted, as it is in part a repetition of the study results.

Author Response

Reviewer reports:

Reviewer 1:

The manuscript is well written but some further corrections or information is necessary to improve the manuscript.

Reply: Thank you for your suggestion.

Abstract:

Line 16: Please add an explanation for the abbreviation RWB in the previous sentence.

Reply: Thank you for your suggestion. We add the explanation for the RWB.

Introduction:

Line 34-38: Please check whether the cited literature is adequate for the statements of the first two sentences.

Reply: In our opinion, this citation is suitable.

Line 49: New machines are available which evaluate the blood smears, but these also have problems, especially with the differentiation of eosinophils in birds.

Reply: Thank you for the information.

Lien 54: Falcons for example have a different morphology of the eosinophils in comparison to other raptor species.

Reply: We rephrase the sentence.

Line 66: Please add: " Which is however not sufficient for the calculation of reference intervals."

Reply: Thank you for your suggestion.

Material and methods:

Line 75: After which time were the blood smears prepared and how long was the transport time of the blood to the laboratory and the start of the blood analysis?

Reply: The rehabilitation center is 200 m far from the laboratory (5 min). Blood smears prepared as soon as possible (when arrive at the laboratory).

Line 84-102: Please add more information about the hematology methods used or add references.

Reply: We add more information about the hematology method.

Line 106: How were the 5 animals selected? Age, sex?

Reply: These 5 animals were selected based on PCV and parasites examination (parasites were not found).

Line 125: How was this cut-off determined?

Reply: Sine the raptor is clinically healthy (physical examination) and there is the RIs for this bird, thus we use this cut-off.

Results:

Line 128-135: The first paragraph of the results should be moved to the Materials and Methods section, as some of this is also a repetition.

Reply: Thank you for your comment. We rephrase the sentence.

Line 170: Lymphocytes with granulation were found in all healthy RWBs?

Figures:

Is it possible to exchange the figure of the lymphocyte against one without granulation or is the granulation typical in this avian species?

Reply: We would like to explain that the lymphocytes with azurophilic granules is not a common circulating lymphocyte but this can be found in healthy RWB. Sine the morphology of lymphocyte is not quite similar to those in other raptors, we would like to show lymphocytes containing azurophilic granules for identification.

Discussion:

Line 212: Please include more other potential reasons for anemia.

Reply: Thank you for your comment. Other potential causes are added.

Line 242-250: The last paragraph of the discussion could be shorted, as it is in part a repetition of the study results.

Reply: Thank you for your comment. I was edited.

Reviewer 2 Report

The article is well structured and clear.  The significance of the results are more difficult to interpret, but this may be a consequence of awkward English and sentence construction.  While this is intended as a baseline article, it still should be made more clear why the results presented here are notable.  The incidence of parasitism is low, but few details were given on how this was determined or the species identified

Author Response

Reviewer 2:

The article is well structured and clear.  The significance of the results are more difficult to interpret, but this may be a consequence of awkward English and sentence construction.  While this is intended as a baseline article, it still should be made more clear why the results presented here are notable.  The incidence of parasitism is low, but few details were given on how this was determined or the species identified

Reply:

We would like to say thank you for your suggestion. The answers are as following:

  1. The significance of the results are more difficult to interpret, but this may be a consequence of awkward English and sentence construction.

= Thank you for your suggestion. English and sentence construction are checked.

  1. .  While this is intended as a baseline article, it still should be made more clear why the results presented here are notable.

= Since the KURRU employ the manual laboratory for analysis of CBC. We need the scientific information for identification of blood cells during the step of WBC differential count. Also, we need the cage mage hematologic value for RWBs.

  1. The incidence of parasitism is low, but few details were given on how this was determined or the species identified

= More detail about parasite identification is available on the previous article as we mention in Materials and Methods.

Reviewer 3 Report

Dear authors,

the topic is very inetersting to me but some things should be corrected.

Do you know animal sex? aproximate age - adults, juvenils? why were they brought to the rescue center - anamnesis? was blood collected from healthy animals? is september a breeding period for buzzards in your country?

also, english lenguage should be improved

78 two species of blood parasites....
112....abbrev - explain
132 two non-parasitized samples showed
135 excluded
135 two non-parasitized with low PCV
142 was
143 TRBS cout was
144 the same
145 and ranged...
155 contained
172 irr-sh nucleus
173 in shape
174 sometimes? how many samples or in what percentage?
177 were shown at the end
196 anemic
210 rephrase
212 perhapse
216 is
217 report
221 is
223 is suggesting or withoit authors
236 as some....
240 sometimes showing
243 was

Author Response

Reviewer 3:

the topic is very inetersting to me but some things should be corrected.

Do you know animal sex? aproximate age - adults, juvenils? why were they brought to the rescue center - anamnesis? was blood collected from healthy animals? is september a breeding period for buzzards in your country?

Reply: Thank you for being the reviewer. We only have the information of age and localities.

also, english lenguage should be improved

Reply: Thank you for your suggestion, we ask for English proofreading.

78 two species of blood parasites....

Reply: Thank you for your suggestion

112....abbrev – explain

Reply: The word “transmission electron microscopy” was added.

132 two non-parasitized samples showed

Reply: Thank you for your suggestion.
135 excluded

Reply: Thank you for your suggestion.

135 two non-parasitized with low PCV

Reply: Thank you for your suggestion.
142 was

Reply: Thank you for your suggestion.
143 TRBS cout was

Reply: Thank you for your suggestion.
144 the same

Reply: We would like to apologize. We don’t see any word relating to your comment.
145 and ranged...

Reply: Thank you for your suggestion.
155 contained

Reply: Thank you for your suggestion.
172 irr-sh nucleus

Reply: Thank you for your suggestion.

173 in shape

Reply: Thank you for your suggestion.

174 sometimes? how many samples or in what percentage?

Reply: The sentence was change
177 were shown at the end

Reply: The sentence was change

196 anemic

Reply: Thank you for your suggestion.

210 rephrase

Reply: We change the sentence "There were 2 non-parasitized RWBs suffered by anemia (PCV < 0.30 L/L) into “One anemic non-parasitized RWBs (PCV < 0.18 L/L) showed hypochromic erythrocytes (Figure 1)”.

212 perhapse

Reply: Thank you for your suggestion.

216 is

Reply: Thank you for your suggestion.

217 report

Reply: Thank you for your suggestion.

221 is

Reply: Thank you for your suggestion.

223 is suggesting or withoit authors

Reply: This is the suggestion for the further investigation.

236 as some....

Reply: We change the sentence “This was similar to those in nocturnal raptors[1], except for some species.” into “This was similar to those in nocturnal raptors[1], except in barn owls and great horned owls”

240 sometimes showing

Reply: Thank you for your suggestion.

243 was

Reply: Thank you for your suggestion
